# Enrichment of Wheat Bread with *Platycodon grandiflorus* Root (PGR) Flour: Rheological Properties and Microstructure of Dough and Physicochemical Characterization of Bread

**DOI:** 10.3390/foods12030580

**Published:** 2023-01-29

**Authors:** Yuanyuan Liu, Qian Zhang, Yuhan Wang, Pingkang Xu, Luya Wang, Lei Liu, Yu Rao

**Affiliations:** 1School of Food Science and Bioengineering, Xihua University, Chengdu 610039, China; 2Chongqing Key Laboratory of Speciality Food Co-Built by Sichuan and Chongqing, Chengdu 610039, China; 3State Key Laboratory of Veterinary Etiological Biology, Chinese Academy of Agricultural Sciences, Lanzhou Veterinary Research Institute, College of Veterinary Medicine, Lanzhou University, Lanzhou 730099, China

**Keywords:** rheological property, dough microstructure, baking performance, flavor attributes, antioxidant property

## Abstract

*Platycodon grandiflorus* (Jacq.) A.DC. root (PGR) flour is well known for its medical and edible values. In order to develop nutritionally fortified products, breads were prepared using wheat flour, partially replaced with PGR flour. The rheological properties and microstructure of dough and the physicochemical characterization of bread were investigated. Results showed that lower level of PGR addition (3 and 6 g/100 g) would improve the baking performance of breads, while the higher level of PGR addition (9 g/100 g) led to smaller specific volume (3.78 mL/g), increased hardness (7.5 ± 1.35 N), and unpalatable mouthfeel (21.8% of resilience and 92.6% of springiness) since its negative effect on the viscoelasticity and microstructure of dough. Moreover, sensory evaluation analysis also showed that the PGR3 and PGR6 breads exhibited a similar flavor to the control bread, but the 9 g/100 g addition of PGR provided bread with an unpleasant odor through its richer volatile components. As expected, the phenolic content and antioxidant capacity of bread increased significantly (*p* < 0.05) as PGR flour was added to the bread formulation. The total phenolic content (TPC) ranged from 14.23 to 22.36 g GAE/g; thus, *DPPH*• and *ABTS*•+ scavenging capacity increased from 10.44 and 10.06 μg Trolox/g to 14.69 and 15.12 μg Trolox/g, respectively. Therefore, our findings emphasized the feasibility of PGR flour partially replacing wheat flour in bread-making systems.

## 1. Introduction

Nowadays, health-conscious consumers increased the demand for innovative and functional food products with nutritious and healthy values. Bread, one of the most important staple foods worldwide, could be used as the vehicle for delivering the nutritional and functional compounds to the consumers [1]. Previously plenty of studies reported that plant’s seeds, leaves, and/or their extracts were introduced to bread and enhanced the nutritional quality of breads [2,3,4]. Especially, fortification of wheat bread with medical plants and herbs obtained the breads with biologically active substances and enhanced healthy quality [5,6].

*Platycodon grandiflorus* (Jacq.) A.DC. root (PGR), commonly applied in medicine and the human diet, is an exceptionally popular and widespread medical herb widely cultivated in Asia, including China, Japan, and Korea [7]. 

PGR has many biological activities, such as anti-cancer, anti-inflammatory, anti-obesity, and anti-allergic functions [8]. Moreover, PGR could also be used in the medical treatment of cough, phlegm, sore throat, and other respiratory disorders [9]. Except for pharmaceutical applications, PGR has a long history of being consumed as food and is popularly recognized for its usage in many dishes, including tea, kimchi, and side dish, and also for the production of alcoholic beverages and preserved fruit [10]. 

Currently, the application of medicinal herbs as a source of antioxidant and nutritional components in bread production has attracted consumers’ attention. Seczyk, Krol, and Kolodziej (2022) [11] developed an innovative bread product with improved antioxidant capacity and potential bio-accessibility using Greek oregano leaves. Moreover, Durovic et al. (2020) [6] comparably investigated the usage of stinging nettle leaves and their extracts in the baking of bread and obtained a bread product with high benefits. However, to our knowledge, there are no studies previously on the application of PGR flour as a bio-functional ingredient applied into bread-making. This is the first time to thoroughly evaluate the influence of PGR flour addition on wheat bread from microstructure level to macroscopically perceptible quality. Rheometer and confocal laser scanning microscopy (CLSM) were used to study the variation in the viscoelasticity and microstructure of dough. Additionally, electronic nose, electronic tongue, and gas chromatography-mass spectrometry (GC-MS) were employed to determine the flavor properties, including taste, odor, and volatile profiles of breads. Furthermore, the baking performances and antioxidant capacity of breads were also investigated. Obtained results of this work can not only develop an innovative and functional bread product fortified with PGR flour but also expand the PGR market to enhance its added value.

## 2. Materials and Methods

### 2.1. Materials

Fresh harvest-cultivated *Platycodon grandiflorus* (Jacq.) A.DC. root (PGR) was purchased from Lao Nong Agricultural Products Co., Ltd., Chifeng, China. The PGR were treated and dried as described by Yuan Yuan Liu et al. (2022) [10]. The dried PGR was converted into homogeneous and fine powder after being ground and sieved through a 100-mesh sieve. Wheat flour, instant dry yeast (Anqi, China), butter, salt, and sugar were purchased from a local supermarket.

### 2.2. Dough Preparation and Bread-Making Process

The doughs and breads were prepared according to a straight-dough method raised by Xu, Luo, Yang, Xiao, and Lu (2019) [12] with slight changes. Formulation was as follows: 200 g of wheat flour, 20 g of butter, 30 g of sugar, 2 g of salt, and 3 g of dry yeast. The wheat flour was replaced by PGR flour at 0, 3, 6, and 9 g/100 g levels, respectively, and the obtained breads were named PGR0 (control), PGR3, PGR6, and PGR9. Above formulation and 90 mL of distilled water were mixed by a professional spiral mixer (MM-ESC1510, Midea, Foshan, China). The first and second mixing procedures lasted 10 min and 30 min, respectively, with an interval of 10 min. Then, the dough was fermented for 60 min that placed in a temperature-humidity-controlled chamber (37 °C, 80% relative humidity). Subsequently, baking was carried out in an oven (DKX-B40R2, Bear, Foshan, China) at an upper temperature of 180 °C and bottom temperature of 210 °C for 30 min. All bread-making processes were conducted in triplicate.

### 2.3. Dynamic Rheological Properties of Dough

The dynamic rheological properties of dough were determined using a rheometer (TA instrument, New Castle, DE, USA) following the method of Y. Cao et al. (2021) [13] with slight modifications. The frequency sweep was conducted at 0.02% of constant strain (within the linear viscosity region), 28 °C of temperature, and 1–100 Hz of frequency. The dough samples without yeast were placed on the 50 mm steel parallel plate with a 2.0 mm gap, and a layer of silicone oil was then employed on the edge of the sample to prevent moisture from evaporating during the experiment. Moreover, the storage modulus (*G*′), loss modulus (*G*″), and loss tangent (tan *δ* = *G*″/*G*′) values were recorded as a function of frequency on the dough samples. Additionally, the obtained *G*′ could be fitted by the power-law model.
(1)$$G^{\prime }\left(\omega \right)=K^{\prime }\omega^{z^{\prime }}$$
where *ω* is the angular frequency (rad/s), *K*′ represents the strength of the dough, and *Z*′ indicates the degree of dependence of *G*′, respectively.

### 2.4. Confocal Laser Scanning Microscopy (CLSM) of Dough

The morphological changes of dough after the addition of PGR were observed through confocal laser scanning microscopy (CLSM) according to the procedure described by Yao et al. (2021) [1] with minor modifications. Briefly, 3–5 mm thickness of dough slices were loaded in slides followed by the stains of 0.25% (*w*/*w*) Fluorescein 5-isothiocyanate (FITC, *λ*_ex_ = 488 nm) and 0.025% (*w*/*w*) Rhodamin B (*λ*_ex_ = 568 nm), which were applied to label the starch and protein fractions, respectively. Bright green and red layers in the CLSM images obtained denoted the starch and protein structures, respectively. 

### 2.5. Bread Basic Quality

#### 2.5.1. Moisture Content, Specific Volume, and Color

The moisture content of the bread was measured using an HB43-S moisture analyzer (Mettler Toledo, Switzerland). The specific volume of the bread was expressed by the ratio of volume (mL) and weight (g), and the volume of the bread was determined by the millet displacement method. 

Meanwhile, the color of the bread crumbs was determined by a spectrometer color analyzer (Lab Scan XE, Hunter Lab, USA). The total difference of color (Δ*E*) was calculated as follows:(2)$$\Delta E=\sqrt{{\left(L-L_{0}\right)}^{2}+{\left(a-a_{0}\right)}^{2}+{\left(b-b_{0}\right)}^{2}}$$
where *L* indicates lightness, *a* indicates the red/green value, and *b* indicates the blue/yellow value. *L*_0_, *a*_0_, and *b*_0_ are the color of PGR0 bread. 

#### 2.5.2. Textural Analysis of Bread Crumbs

The texture of bread crumbs was determined by a TA.XT Plus texture analyzer equipped with a 36 mm cylindrical probe (P/36R). TPA measurement of bread was performed with a pretest speed of 1.0 mm/s, test speed of 1.0 mm/s, post-test speed of 2 mm/s, and deformation level of up to 60%. There was 30 s for the dough balance between the two-time compression. Crumb characteristics of hardness, resilience, cohesion, and springiness were recorded [14]. 

### 2.6. Flavor Properties Analysis

#### 2.6.1. Electronic Nose

The electronic nose (Isenso Group Corporation, New York, NY, USA) was employed to determine the taste of bread samples, which could detect different odors by 14 metal oxide sensor arrays. Before the measurements, 3 g of bread samples were sealed in a headspace bottle and left at 35 °C for 10 h to equilibrate the volatile flavors inside. Then, the electronic nose tests were conducted under the following conditions: 300 s of cleaning time, 120 s of preparation time, 0.6 L/min of flow rate, 25 °C of detection temperature, 55 ± 2% of relative humidity, 60 s of detection time, 100 s of zero count, and 10 s of automatic adjustment zero time. Ultimately, the characteristic values of each sensor were extracted as the electronic nose results [1], and data between 81 and 85 s after stabilization were taken for principal component analysis (PCA).

#### 2.6.2. Electronic Tongue Profiles

The taste properties of PGR bread were determined by an electronic tongue (Smar Tongue, Isenso Group Corporation, USA) as the method proposed by Yuan Yuan Liu et al. (2021) [7]. A total of 0.5 g of PGR bread was extracted in 100 mL of reference solution composed of 0.3 mM tartaric acid and 30 mM KCl at 70 °C for 2 h. Then the supernatant was obtained after centrifugation, and 15 mL was used for the subsequent measurements. All sensors were selected, and the total time for sampling and cleaning was 4 min. The electronic tongue was self-tested prior to measurement, activation, calibration, and diagnostic steps to ensure the reliability and stability of the data collected. The characteristic value of each sensor was extracted for analysis, and measurements were repeated 5 times to verify the effectiveness of the statistical experiment.

#### 2.6.3. Identification of Bread Volatile Compounds 

Gas chromatography-mass spectrometry (GC-MS) was used to further analyze the aroma characteristics between different bread samples, according to Yuan Yuan Liu et al. (2021) [8]. A total of 1 g of bread samples were dehydrated in a hot air oven at 60 °C for 5 min. Then, the samples were added into a 20 mL headspace bottle, incubated at 60 °C for 2 h, and finally injected manually with an HP-5 MS capillary column. GC-MS analysis was conducted by a 7890A/5975 GC/MSD system as the following parameters: HP-5 MS capillary column (30 m × 0.25 mm, 0.25 mm), 250 °C of injection port temperature, 60 °C of oven temperature, 1.0 mL of column flow rate, 280 °C of Aux-2 temperature, 230 °C of ion source temperature, and 150 °C of quadrupole temperature. GC-MS spectra were further analyzed through the National Standards and Technology (NIST05) database and related literature to identify known and unknown volatile compounds in bread.

### 2.7. Antioxidant Properties

The bread extracts were prepared through the method reported by Sagar and Pareek (2021) [2]. That is, 1 g of bread crumbs was extracted with 10 mL of 80% methanol by a rotatory shaker under the condition of 37 °C and 90 rpm for 2 h. Then the mixed solution was centrifuged at 12,000× *g* for 15 min at 4 °C. Above procedures were repeated twice, and the supernatants were collected as the extract solution for subsequent analysis. 

#### 2.7.1. Total Polyphenol Content (TPC)

The total polyphenol content (TPC) determination of breads was carried out by the Folin–Ciocalteu method according to Velioglu, Mazza, Gao, and Oomah (1998) [15] with slight modifications. A total of 1 mL of bread extract was mixed with 2 mL of 12% sodium carbonate solution and 1 mL of Folin–Ciocalteu reagent, followed by incubation for 1 h in the dark. Then, the absorbance of the solution was recorded using a UV- spectrophotometer (TU-1810, Beijing Puekinje General Instrument Co., Ltd., Beijing, China) at 750 nm. The TPC of bread samples was expressed as g gallic acid equivalent (GAE)/g obtained from a calibration curve (y = 11.3800x + 0.0318, *R*^2^ = 0.9940).

#### 2.7.2. DPPH• scavenging activity

*DPPH*• scavenging activity determination of bread samples was carried out. A total of 10 mg of DPPH was dissolved in 25 mL of 80% methanol to obtain a methanolic DPPH solution. Moreover, the blank was prepared by the mixture solution of 2 mL of methanolic DPPH solution and 2 mL of 80% methanol. A total of 2 mL of bread extract was mixed with 2 mL of methanolic DPPH solution. The mixed solution was shaken thoroughly and then placed at room temperature to incubate for 30 min, and the absorbance was recorded at 517 nm. The *DPPH*• scavenging activity of the sample was expressed as μg Trolox/g calculated from a calibration curve (y = −19.45x + 0.588, *R*^2^ = 0.9940).

#### 2.7.3. ABTS•+ Scavenging Activity

*ABTS*•+ scavenging activity of bread was determined using the methods reported by Re et al. (1999) [16] with slight modifications. *ABTS*•+ stock solution was prepared with 7 mM *ABTS* solution and 2.45 mM potassium persulphate after being placed in darkness for 12 h. Then, the *ABTS*•+ stock solution was diluted with distilled water until the absorbance of 0.7 ± 0.02 at 734 nm to obtain an *ABTS*•+ working solution. A total of 0.2 mL of bread extract was added to 0.8 mL *ABTS*•+ working solution to incubate for 6 min. Then, the absorbance was measured at 734 nm, and the *ABTS*•+ scavenging activity was expressed as μg Trolox/g calculated from a calibration curve (y = −24.07x + 0.5496, *R*^2^ = 0.9981).

### 2.8. Statistical Analysis

All tests were carried out at least in triplicate at least, and the data were expressed as means ± standard deviation. Statistical analysis was conducted by one-way analysis of variance (ANOVA) followed by Duncan’s multiple range test using SPSS 17.0 (SPSS Inc., Chicago, IL, USA) at a significant level of *p* ≤ 0.05. Orthogonal partial least squares-discriminant analysis (OPLS-DA) was conducted by Simca 14.1 (Umetrics, Umea, Sweden) to calculate the variable influence on projection (VIP) values. The principal component analysis (PCA) and discriminant factor analysis (DFA) were carried out using Origin 2021 software (Origin Lab Corporation, Northampton, MA, USA).

## 3. Results and Discussion

### 3.1. Rheological Properties

Analysis of rheological properties could reflect the influence of recipe components and/or technology modifications on the dough properties especially viscoelastic characteristics [17], which had a relationship with the quality of resultant breads from the dough, such as load volume and crumb cell structure [18]. Appendix A illustrated the storage modulus (*G*′) and loss modulus (*G*″) values of five doughs with different concentrations of PGR powder from 0 g/100 g to 9 g/100 g. The *G*′ values of all five doughs were higher than their corresponding *G*″ values, indicating the viscoelastic characteristics and solid-like behavior of PGR doughs [17,19]. Additionally, the addition of PGR powder affected the rheological properties of the wheat dough; that is, as the concentration of PGR powder in the dough increased, the *G*′ and *G*″ values both exhibited descending trend firstly, then ascending trend in the frequency range from 1 to 100 Hz. Replacing 3 g/100 g of wheat flour with PGR powder in the dough system resulted in a significant decrease in *G*′ and *G*″ values, which could be attributed to the negative effect of PGR flour, rich in fiber, on the structure and intensity of gluten network [20,21]. Guo et al. (2022) [19] also pointed out that the hydrogen bonds were easily formed between PGR and water molecules, causing a decrease in gluten content and the weakness of the gluten network. While the increased trend of *G*′ and *G*″ values showed by the dough with 6 g/100 g and 9 g/100 g of PGR powder could be ascribed either to the lack of water lubrication caused by the competition for water between gluten and fiber of PGR or to the role of fiber as fillers in the viscoelastic matrix [22]. Such an increasing trend was also observed in previous studies when wheat dough was added to wheat bran [23]. Simultaneously, Van Bockstaele, De Leyn, Eeckhout, and Dewettinck (2008) [24] also found that there is an inverse correlation of loaf volume to dough dynamic rheological parameters including *G*′ and *G*″. A higher rheological modulus suggests that more gas pressure was required to expand the dough during proofing [25] so that the dough prepared with PGR flour might lead to a lower bread volume.

The loss tangent (tan *δ*) was the ratio of *G*″ and *G*′, which could be used to evaluate the protein quality [19]. As shown in Appendix A, the tan *δ* values of all doughs were in the range of 0.1 to 1, suggesting the solid-like behavior and weak-gels characteristics of the analyzed doughs [17]. Compared with control dough (PGR0), the tan *δ* of dough containing 6 g/100 g and 9 g/100 g of PGR powder decreased in the frequency ranged from 10 to 100 Hz, suggesting that the prepared dough with 6 g/100 g and 9 g/100 g PGR powder was more elastic and rigid. Moreover, the power-law model fitting results are exhibited in Table 1. The *Z*′ values indicate the degree of dependence of *G*′, and the nature of molecular interaction in dough [26] and *Z*′ > 0 revealed a less stable network structure formed by a physical linkage [27]. As shown in Table 1, the *Z*′ values of prepared doughs with different addition of PGR flour ranged from 0.220 to 0.303, suggesting the low-stability physical linkage of the dough network structure. Furthermore, the decreased *Z*′ values as the increased addition of PGR powder also indicated that the dough becomes less time-dependent, which structure resembles that of a highly cross-linked material [25], resulting in the formation of the linkages between PGR and wheat proteins. Additionally, the *K*′ value refers to the strength of the dough matrix higher *K*′ values mean more strengthened dough [14]. The *K*′ values of the prepared dough with 0 g/100 g, 3 g/100 g, 6 g/100 g, and 9 g/100 g PGR flour were 5.078, 5.053, 6.020, and 8.115 Pa∙s^z′^, respectively. As a whole, the incorporation of PGR powder into wheat doughs decreased *Z*′ values but increased *K*′ values of prepared doughs, suggesting the enhanced consistency and strength of wheat doughs. According to Sun et al. (2022) [18], the strength of the dough was an important factor in determining the quality of the resultant bread. Doughs that were too strong were hard to develop bubbles appropriately, leading to small, dense, and unpalatable loaves. Inversely, too-weak dough cannot hold the bubbles and will lead to the formation of large holes or the collapse of the loaf. 

### 3.2. Dough Microstructure Analysis

CLSM was employed to observe the microstructure of the dough prepared by different formulations, and the results are shown in Figure 1. The starch granules are stained in green, but gluten proteins are marked in red. Meanwhile, the black part might be the pores, water, or other substances [1]. The gluten network structure of the model wheat dough (Figure 1(A-1–A-3)) was relatively uniform, continuous, and compact, and the gluten protein was embedded in the starch matrix. In addition, the dough with added 3 g/100 g PGR flour exhibited a denser gluten network, and there was a few gas cells in the gluten structure (Figure 1(B-1–B-3)). This could be attributed to the high water-holding capacity of PGR flour, which could be embedded in the gluten network. Additionally, the fiber in the PGR powder interacted with the gluten via hydrogen bonds and played the role of filler in the network structure, ultimately causing the enhancement of the gluten structure in the dough [28]. Similarly, fiber nanoparticles in PGR flour could lead to an increase in surface area and limited plasticization effect, which also help to improve the strength of PGR dough. In the case of 6 g/100 g PGR and 9 g/100 g PGR (Figure 1(C-1–C-3,D-1–D-3)), it is worth noting that the continuous protein network is less pronounced, and the gluten protein network is disrupted by more and large gas cells. Especially the dough with 9 g/100 g PGR exhibited a discontinuous gluten network structure (Figure 1(D-1–D-3)). Due to the impaired strength of gluten filaments and diluted gluten protein by high level of PGR flour in dough, there was a looser gluten network structure with more gas cells and destroyed gluten-starch matrix [29]. A comparable change in the microstructure information of wheat dough through the addition of berry pomace has been reported by Struck et al. (2018) [25]. It is also pointed out that the excessive application of fiber in the dough would restrict the gluten and lower the elasticity and extensibility of dough, resulting in lower gas retention and, therefore, smaller loaf volume (Figure 1(D-4)). Consequently, as shown in Figure 1(A-4–D-4), due to the distinction of gluten network in the dough prepared by different addition of PGR, a visible difference could be observed in final bread. Therefore, the 3 g/100 g of incorporation seems promising for the successful application of PGR in wheat bread.

### 3.3. Bread Basic Quality 

#### 3.3.1. Specific Volume, Moisture Content, and Color

Among various physical attributes of bread, specific volume, color, and texture properties were considered as a proxy for consumer acceptance. High bread quality usually has the characteristics of higher specific volume, cohesive palatable, and soft crumb texture [14]. As can be seen from Table 2, the specific volume of bread was related to the level of PGR addition; that is, the specific volume of bread first increased from 5.1 mL/g (bread with 0 g/100 g PGR flour) to 5.1 and 5.4 mL/g (bread with 3 g/100 g and 6 g/100 g PGR flour). This improved effect could be attributed to the suitable gluten structure in PGR3- and PGR6-bread, which was previously confirmed by the enhanced strength and dough microstructure, as indicated in Section 3.1 and Section 3.2, ultimately resulting in better gas holding capacity and higher specific volume. While the bread with 9 g/100 g PGR powder showed the lowest specific volume (3.8 mL/g), suggesting the inferior fermentation properties of dough with 9 g/100 g PGR flour. This could be due to the weakening dough structure resulting from the water competition between PGR fiber and gluten. Previous literature reported that fiber-rich constituents such as okra and quinoa flour added to wheat bread caused a significant reduction in loaf volume [3,12]. Similarly, the rheological properties and CLSM results (Section 3.1 and Section 3.2) indicated that replacing wheat flour with PGR powder would not only reduce the amount of gluten but affect the properties of gluten. That is, a 9 g/100 g addition of PGR would significantly increase the strength of the gluten network, and over strong gluten, the network limited the expansion of dough during the fermentation and proofing stage. Meanwhile, the dough with 9 g/100 g PGR flour showed a fibrous gluten network with numerous holes so that it was unable to retain CO_2_ gas, causing its lower specific volume. The moisture content of bread would affect its shelf life and determine the ease of deterioration. The addition of PGR caused the changes in crumb moisture content (Table 2), which firstly increased from 38.9% (PGR0 bread) to 39.5% (PGR3 bread) and then decreased to 37.6% and 37.9% (PGR6 bread and PGR9 bread, respectively). Wójcik et al. (2021) [5] previously reported that the moisture content of bread showed a significantly negative correlation with bread volume. However, in the case of our study, no significant correlation was observed in bread volume versus moisture content (*r* = 0.166, *p* = 0.607). This inconsistence might be attributed to the difference in the bread-making process that Wójcik et al. (2021) [5] employed a method based on scalded flour. 

Color is another critical evaluation of bread quality that is closely related to the consumer’s purchasing decision [7], and bread color is mainly determined by factors including formulation and baking conditions. Total color difference (Δ*E*) represents the magnitude of the color difference between different bread samples. The Δ*E* value of bread added with PGR powder ranged from 4.9 to 5.8 (>3), indicating that there is a perceivable color difference between breads enriched with PGR powder compared with pure wheat bread (PGR0). Furthermore, the lightness (*L*) of four breads showed significant differences that *L* values are 55.8 ± 2.08 for PGR0 bread, 60.0 ± 1.24 for PGR3 bread, 61.4 ± 2.63 for PGR6 bread, and 54.5 ± 3.76 for PGR9 bread. The significantly lower *L* value of PGR9 bread than that of PGR3 and PGR6 breads could be attributed to the fact that the 9 g/100 g addition of PGR flour in bread formulation would provide more precursors for the Maillard reaction during baking that weaken the enhanced effect of PGR addition on the *L* values of breads. Likewise, the redness (*a*) and yellowness (*b*) values of four bread samples exhibited significant differences, and the *a* and *b* values both increased with the addition of PGR. The highest *a* and *b* values were both found for the bread with 9 g/100 g PGR (2.3 and 21.1 for *a* and *b* values, respectively) and followed by the PGR6 bread (*a* and *b* values were 1.9 and 20.2). Higher redness of the product would enhance its visual attraction, and more yellowness was preferred in PGR-related products [7,30]. Overall, regarding the *a* and *b* parameters of bread samples, 6 g/100 g and 9 g/100 g of PGR addition in the bread-making formulation was more desirable.

#### 3.3.2. Texture Profile Analysis (TPA)

Texture properties are close to the mouthfeel of bread, which are important indicators to evaluate the consumers’ preference for bread [13]. As seen in Table 2, the inclusion of PGR flours led to changed crumb hardness, which was inversely correlated with the specific volume. Thus, PGR9 bread exhibited the lowest loaf volume (3.8 ± 0.78 mL/g) but the highest hardness (7.5 ± 1.35 N). Such an inverse correlation of bread-specific volume with hardness was also reported in previous studies [3,5,12]. As discussed in Section 3.1, PGR-9 dough exhibited a significantly higher *K*’ value than the other three doughs, indicating that dough with 9 g/100 g PGR flour had a stronger gluten network. Moreover, the stronger gluten structure would limit the expansion of dough during fermentation, resulting in lower specific volume and harder texture of bread [13]. Additionally, Tebben et al. (2018) [31] indicated that the crumb moisture content was also a critical factor affecting the bread firmness and loaf-specific volume; that is, higher moisture content would decrease the hardness of bread. Therefore, the bread prepared with a 3 g/100 g addition of PGR, which had the highest moisture content, exhibited a relatively softer texture than PGR0 bread. 

As shown in Table 2, the bread added with PGR flour exhibited apparently lower resilience compared with the pure wheat bread (PGR0), whereas significantly decreased springiness and cohesiveness were only observed in the PGR9 bread. Generally, the increased hardness of bread contributed to the decreased crumb elasticity [32]. Therefore, PGR9 bread that has the highest hardness (7.5 ± 1.35 N), showed the lowest resilience (21.8 ± 1.22%) and springiness (92.6 ± 1.57%). On the other hand, PGR flour contains relatively less gluten protein than that wheat flour, resulting in lower stability of gluten network structure as the PGR flour content increased in bread formulation. Also discussed in 3.2, PGR9 dough showed a discontinuous and rough cross-linked gluten network with irregular holes. Therefore, the high level of PGR flour added to the dough disrupted the gluten network and caused a discontinuous or even fibrous gluten network, leading to leading to the apparently reduced resilience, springiness, and cohesiveness of the final bread. 

Finally, based on the texture attributes of PGR enriched breads, PGR3 and PGR9 breads that exhibited relatively higher specific volume, satisfied color, softer crumb, and palatable texture are the closest in appearance to the control wheat bread (PGR0) and more likely to obtain consumers’ preference. 

#### 3.3.3. Orthogonal Partial Least Squares-Discriminant Analysis (OPLS-DA) Analysis

The OPLS-DA model is able to filter out variables that are not relevant to the subgroup by decomposing the *X*-axis matrix information into two types of information that are relevant or irrelevant to Y. This, combined with the variable influence on projection (VIP) value of the variability variables, results in a more reliable variability indicator being obtained. OPLS-DA model was performed by adding bread basic properties and evaluation of breads to each group. As seen in Figure 2A, the independent variable of R^2^X (cum) = 0.838, R^2^Y (cum) = 0.549, and Q2 (cum) = 0.345 indicated a reliable predictive ability and stability of the model. Moreover, four bread samples population were located within the 95% confidence interval and exhibited an obvious aggregation tendency, achieving better separation. Variable importance in the projection (VIP) could be used to determine the contribution of variables to the OPLS-DA prediction model, and VIP > 1 suggested a significant contribution. Therefore, the color, texture properties except for resilience and springiness, and specific volume were considered as the contribution indicators of PGR bread (Figure 2B), and there was a significant difference in these indicators between the four bread samples. Additionally, the loading plot diagram (Figure 2C) represented the different correlations between the four breads and their basic properties, and the distances between indicators to coordinate the center point suggested their contribution to the difference between samples. Therefore, Figure 2C further supports the results of the VIP value.

### 3.4. Bread Flavor Properties

#### 3.4.1. Electronic Tongue Profiles

Electronic tongue and nose were employed to investigate the flavor quality of bread samples. PCA analysis is the data transformation and dimensionality reduction of the extracted information from multiple indicators of the sensor and the linear classification of the reduced feature vectors. Moreover, DFA analysis is a further optimization of the response signal data for taste based on PCA to maximize the variability of the data so that bread samples of different tastes can be better differentiated. The discriminant index (DI) (ranged from −100 to 100) of PCA and DFA represented the degree of difference among samples, and lower DI values indicated a higher similarity. Figure 3(A-1,A-2) displayed the analysis results of PCA and DFA for the different breads by the electronic tongue, with the respective contribution of PC1 and PC2 of 59.28% and 13.50%. DI values were 89.17 (PCA) and 97.80 (DFA), respectively. PGR3 and PGR6 breads showed a similar taste on PC1 and PC2, whereas they presented significantly different from the control bread (PGR0) of the PC2 axis. Notably, PGR9 bread exhibited a significant difference in sensors of electronic tongue lies on the PC1 axis and PC2 axis from PGR0 bread, indicating that the 9 g/100 g addition of PGR flour in the bread formulation destroyed the original taste of wheat bread.

#### 3.4.2. Electronic Nose Analysis

The aroma of breads, which is generally evaluated by the electronic nose, is a key factor that determines the acceptance of consumers, and the ingredients of the recipe strongly affect the final aroma [33]. As shown in Figure 3(B-1,B-2), the respective contribution of PC1 and PC2 of 88.3% and 8.0%, and the cumulative contribution rate was 91.3%. Moreover, the DI value was 95.45 (PCA) and 99.87 (DFA), indicating that PGR flour addition significantly changed the aroma quality of bread. Moreover, similar to the electronic tongue results, the most apparent difference in bread aroma was also presented between PGR9 bread and PGR0 bread samples. Figure 3(B-3) exhibited the radar distribution results of aroma analysis, and 14 flavors were detected in the 4 bread samples. Among that, the VOCs (S7) were more apparent. Furthermore, the aroma of PGR6 and PGR9 was more abundant than that of PGR0 and PGR3 bread samples, and the content of flavor substance in PGR0 and PGR3 breads showed little difference. Notably, PGR9 bread presented more unpreferred odors, including sulfide (S1), alcohol (S3 and S11), and smoking odor (S12), which would negatively affect the acceptability of consumers. 

#### 3.4.3. Volatile Profiles

To further investigate the detailed compounds that contribute to the sensory properties of bread samples, the volatile profiles in four bread samples were analyzed by GC-MS. The rich and unique aroma of bread is mainly formed by abundant volatile compounds such as alcohols, aldehydes, esters, ketones, and acids derived from the Maillard reaction [34]. A total of 16 volatile components were detected in tested breads (Figure 4), including 5 kinds of alkenes, 4 kinds of ketones, 3 kinds of alcohols, 2 kinds of esters, and 2 kinds of aldehydes. Of these identified volatile components, 9 volatiles were considered to be related to the bread aroma, including 1-hexanol, 1-heptyn-3-ol, 6-methyl-3,5-heptadiene-2-one, carvone, ethyl octanoate, furfural, copaene, and trans-alpha-bergamotene. 

Expectedly, the types of volatile compounds in tested breads were similar but varied in concentration. The most abundant variety of volatile compounds was found in the PGR9 bread. Alcohol is mainly derived from microbial metabolism. 1-Hexanol was only detected in the PGR3 bread, and its content was 0.3 μg/g. According to Joana Pico, Bernal, and Gómez (2015) [35], 1-Hexanol would provide the bread green grass, woody, and flowery aroma notes. Additionally, the content of 1-heptyn-3-ol and phenylethyl alcohol in PGR0 bread was 3.99 μg/g and 1.48 μg/g, which were higher than that of PGR3 bread (2.5 μg/g and 1.1 μg/g) and PGR6 bread (1.04 and 0.84 μg/g), but lower than that of PGR9 bread (4.72 and 3.8 μg/g). Both 1-heptyn-3-ol and phenylethyl alcohol were considered as higher alcohols that were the main primary metabolites formed by *S. cerevisiae* and led to a fuller and soft taste of the product [36]. Aldehydes could provide fermented foods with a mellow and pleasant flavor. Particularly, furfural was considered important for the bread odor quality, which exhibited an almond, caramel, and toasted odor for the bread and was primarily come from the Maillard reaction and 1,2-enolisation during the bread process [35]. PGR flour is mainly composed of carbohydrate (49.8 g/100 g) and protein (5.47 g/100 g), which could provide carbonyl compounds and carbonyl compounds for the Maillard reaction occurring during the baking process, ultimately contributing to the formation of color, flavor/off flavor, or volatile compounds of breads. Similar to alcohol results, the most content of furfural was found in PGR9 bread, while the least furfural was seen in PGR3 bread. Moreover, a unique aldehyde that is 4-ethoxy-3-anisaldehyde was only detected in PGR6 bread. Ketones were mainly produced by the decomposition of esters and alcohols. As well, some ketones are associated with special aromas, such as 6-methyl-3,5-heptadiene-2-one (herbal, fresh, citrus notes) and carvone (caraway/dill, sweet spearmint notes). Furthermore, PGR9 bread also had the highest total content of ketones, followed by PGR0, PGR6, and PGR3. Esters usually come from a combination of low-grade saturated fatty acids and alcohols [37]. Two esters, including ethyl octanoate and methyl 2-methylvalerate, were detected. Ethyl octanoate with fruit aroma showed a relatively higher content (>2 μg/g) than other volatile components in four bread samples. Additionally, alkenes also contributed importantly to the aroma of bread. Copaene with tomato aroma showed the significant higher content in PGR9 bread, while the highest content of trans-alpha-bergamotene with wood and tea order were found in PGR0 bread. Additionally, spiro[5.5]undec-2-ene, 3,7,7-trimethyl-11-methylene-, (-)- were only measured in PGR6 and PGR9 breads, indicating that the enough addition of PGR powder into dough formulation is needed for the generation of spiro[5.5]undec-2-ene, 3,7,7-trimethyl-11-methylene-, (-)-. 

In order to better describe the similarities and differences in the flavors formed by the volatile quantified compounds in the tested breads, a hierarchical clustering analysis was carried out. As seen in Figure 4, the volatile components in bread can be divided into the following categories: (I) trans-alpha-bergamotene, ethyl octanoate, copaene, phenylethyl alcohol; (II) santolina triene, 1-heptyn-3-ol, and isoborneol; (III) methyl 2-methylvalerate, furfural, and 4-Oxatricyclo[4.3.1.1(3,8)]undecan-5-one; (IV) 4-ethoxy-3-anisaldehyde, 1-hexanol, carvone, spiro[5.5]undec-2-ene, 3,7,7-trimethyl-11-methylene-, 4H-Pyran-4-one, 2,3-dihydro-3,5-dihydroxy-6-methyl-, and 6-methyl-3,5-heptadiene-2-one. Moreover, PGR9 bread exhibited higher amounts of I volatile compounds than the other three breads. In contrast, the amount of II volatile compounds in PGR0 bread is higher than in other breads with the addition of PGR flour. Furthermore, also according to Figure 4, PGR3 and PGR6 were separated from the others and clustered together with the PGR0, indicating that there was little significant difference in the aroma profiles among PGR0, PGR3, and PGR6 breads.

### 3.5. Antioxidant Properties Analysis

In order to explore the effect of PGR flour addition on the antioxidant properties of breads, the total polyphenol content (TPC), *DPPH*•, and *ABTS*•+ scavenging capacity were determined, and the results were seen in Figure 5. The addition of PGR flour into bread formulation positively improved the TPC values of bread, and PGR9 bread showed the highest TPC value, followed by PGR6, PGR3, and PGR0 breads. Additionally, the breads with PGR flour exhibited significantly stronger antioxidant activities than the control bread (PGR0), which was expected because the antioxidant capacity of food products was strongly associated with their polyphenol content [12]. *DPPH*• and *ABTS*•+ scavenging activities were usually used as indicators of antioxidant capacity. As shown in Figure 5, compared with PGR0 bread, PGR9 bread showed a 1.41-fold increase in *DPPH*• scavenging capacity and a 1.50-fold increase in *ABTS*•+ scavenging capacity. Similar results were previously reported by Sagar and Pareek (2021) [2] and X. Xu et al. (2019) [12], in which the addition of onion skin powder and quinoa flour both increased the antioxidant activities of wheat bread. Consequently, the fortification of wheat flour with PGR powder contributed to the nutraceutical potential of wheat bread.

## 4. Conclusions

In order to develop a new functional product with improved baking performance and biochemical quality, PGR flour was added to the bread. Dough characteristics, including rheological properties and dough microstructure, were apparently affected by the addition of PGR flour. Consequently, the baking performance of obtained breads was varied. For the results, it is noteworthy that the addition of a high level of PGR flour (9 g/100 g) had a negative effect on the quality of bread, such as lower specific volume and inferior texture attributes. While the addition of 3 g/100 g and 6 g/100 g PGR flour maintained or improved the bread quality. Besides the baking performance, the biochemical quality of the bread was also evaluated. Results showed that PGR0 and PGR3 bread was closer in flavor properties. Nevertheless, the 9 g/100 g addition of PGR flour significantly changed the flavor of bread through richer volatile components. Notably, the antioxidant capacity of wheat bread was improved as the addition of PGR flour increased. Therefore, considering the acceptance of consumers, it could be concluded that the 3 g/100 g and 6 g/100 g addition of PGR flour successfully substituted wheat flour in bread, and these results provided a thought to future research about the use of PGR in the bakery industry.

## Figures and Tables

**Figure 1 foods-12-00580-f001:**
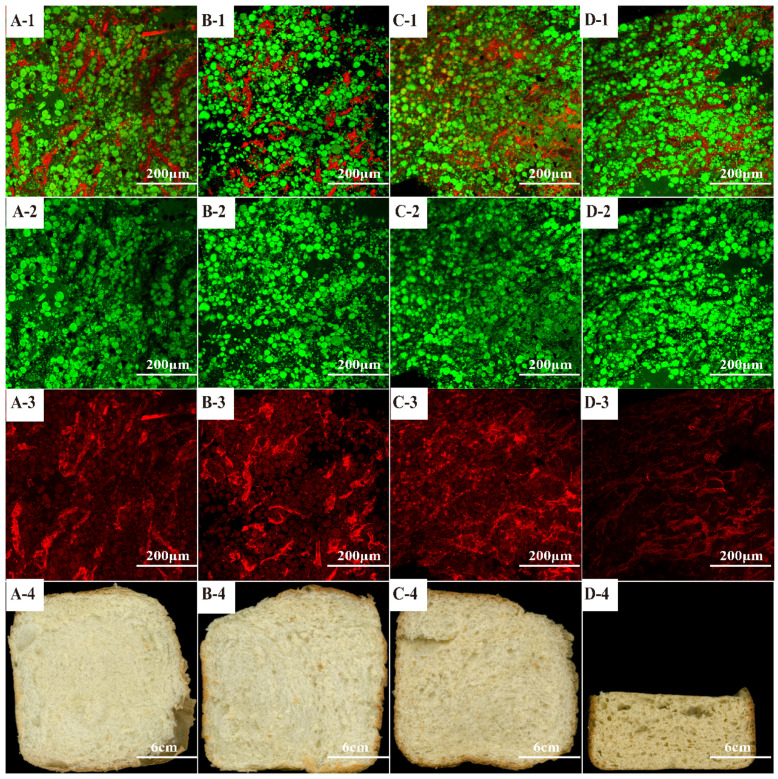
CLSM micrographs of dough matrix and bread pictures. (**A1**–**A4**): PGR0; (**B1**–**B4**): PGR3; (**C1**–**C4**): PGR6; (**D1**–**D4**): PGR9. (**A**–**C**,**D**-**1**–**D**-**3**): CLSM micrographs; (**A**–**C**,**D**−**4**): Pictures of bread slices.

**Figure 2 foods-12-00580-f002:**
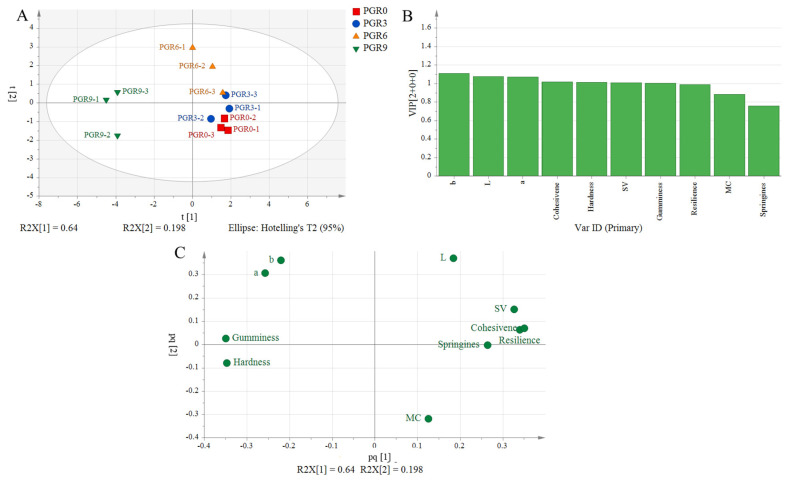
OPLS-DA analysis result. PGR0: without PGR flour; PGR3: with 3 g/100 g PGR flour; PGR6: with 6 g/100 g PGR flour; PGR9: with 9 g/100 g PGR flour. (**A**): OPLS-DA score plot; (**B**): VIP; (**C**): Loading plot. SV: specific volume; MC: moisture content.

**Figure 3 foods-12-00580-f003:**
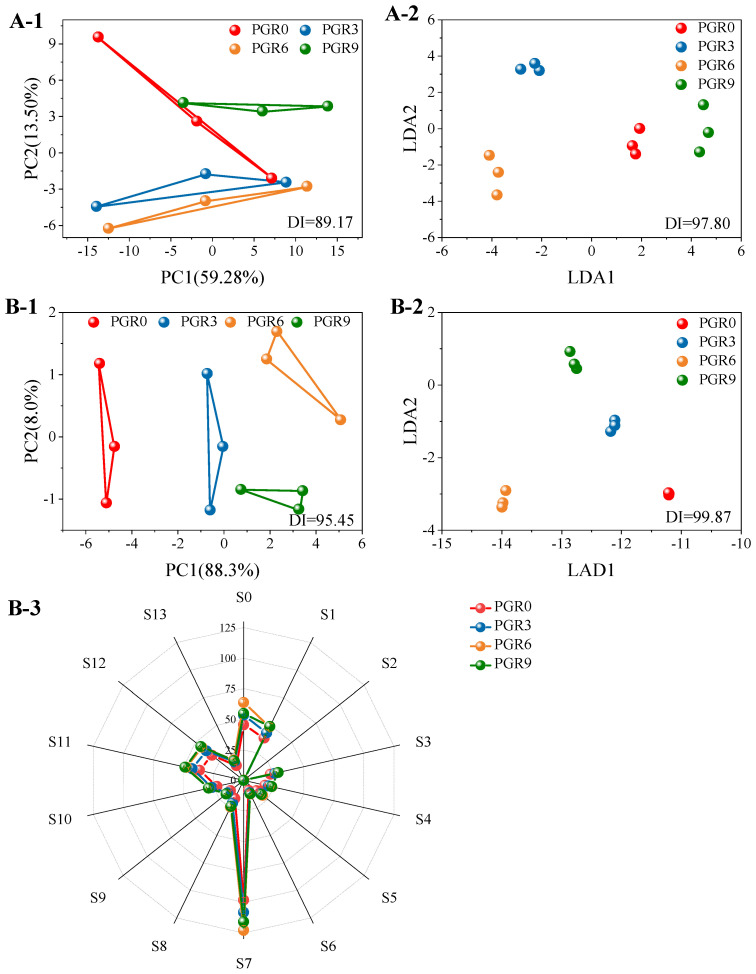
Electronic tongue and electronic nose results of wheat bread enriched with different addition amounts of PGR flour. (**A**-**1**,**A**-**2**): PCA and DFA analysis of electronic tongue. (**B**-**1**–**B**-**3**): PCA, DFA analysis, and spider plot of an electronic nose. PGR0: without PGR flour; PGR3: with 3 g/100 g PGR flour; PGR6: with 6 g/100 g PGR flour; PGR9: with 9 g/100 g PGR flour.

**Figure 4 foods-12-00580-f004:**
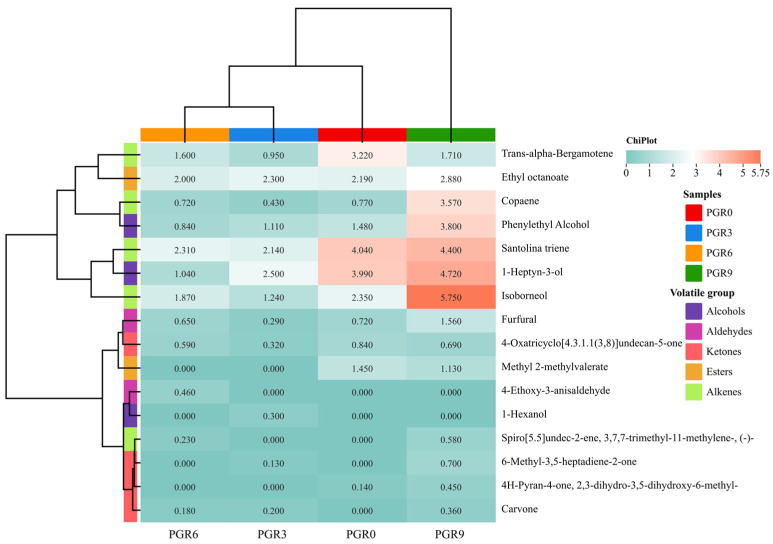
Volatile profiles of wheat bread enriched with different addition amounts of PGR flour. PGR0: without PGR flour; PGR3: with 3 g/100 g PGR flour; PGR6: with 6 g/100 g PGR flour; PGR9: with 9 g/100 g PGR flour.

**Figure 5 foods-12-00580-f005:**
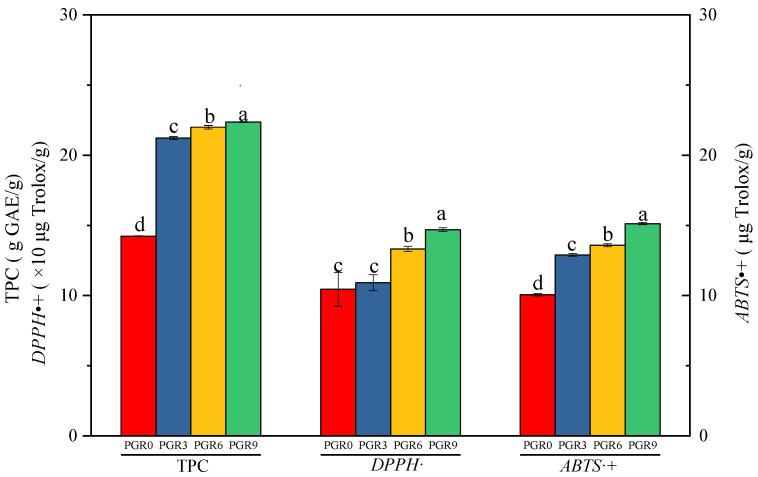
Total polyphenol content (TPC) and antioxidant capacity (DPPH·and ABTS·+) of wheat bread enriched with different addition amounts of PGR flour. PGR0: without PGR flour; PGR3: with 3 g/100 g PGR flour; PGR6: with 6 g/100 g PGR flour; PGR9: with 9 g/100 g PGR flour. Values with different letters are significantly different (*p* ≤ 0.05).

**Table 1 foods-12-00580-t001:** Parameters of the functions describing viscoelastic properties of dough with different addition amounts of PGR flour.

	*K*′ (Pa·s^z′^)	*Z*′ (Dimensionless)	*R* ^2^
PGR0	5.078 ± 0.168 ᶜ	0.303 ± 0.009 ᵃ	0.984
PGR3	5.053 ± 0.044 ᶜ	0.220 ± 0.003 ᶜ	0.997
PGR6	6.020 ± 0.034 ᵇ	0.246 ± 0.002 ᶜ	0.999
PGR9	8.115 ± 0.135 ᵃ	0.284 ± 0.005 ᵇ	0.995

^a–c^ Values in the same row with different superscript letters differ significantly (*p* ≤ 0.05). PGR0: without
PGR flour; PGR3: with 3 g/100 g PGR flour; PGR6: with 6 g/100 g PGR flour; PGR9: with 9 g/100 g PGR
flour.

**Table 2 foods-12-00580-t002:** Basic quality of wheat bread enriched with different addition amounts of PGR flour.

		PGR0	PGR3	PGR6	PGR9
Specific volume (mL/g)		5.1 ± 0.42 ^a^	5.1±0.50 ^a^	5.4 ± 0.18 ^a^	3.8 ± 0.78 ^b^
Moisture content (%)		38.9 ± 0.47 ^ab^	39.5±0.71 ^a^	37.6 ± 0.39 ^b^	37.9 ± 1.74 ^ab^
Color	*L*	55.8 ± 2.08 ^b^	60.0±1.24 ^a^	61.4 ± 2.63 ^a^	54.5 ± 3.76 ^b^
	*a*	1.0 ± 0.13 ^b^	1.3±0.33 ^b^	1.9 ± 0.32 ^a^	2.3 ± 0.43 ^a^
	*b*	17.9 ± 0.82 ^c^	18.6±1.00 ^bc^	20.2 ± 1.57 ^ab^	21.1 ± 1.13 ^a^
	Δ*E*	−	4.9±0.31 ^a^	5.2 ± 0.21 ^a^	5.8 ± 0.70 ^a^
Textural characteristic	Hardness(N)	1.7 ± 0.12 ^b^	1.3±0.24 ^b^	1.7 ± 0.05 ^b^	7.5 ± 1.35 ^a^
	Resilience(%)	33.3 ± 1.11 ^a^	30.9 ± 1.48 ^b^	31.3 ± 0.91 ^b^	21.8 ± 1.22 ^c^
	Cohesiveness (%)	0.8 ± 0.03 ^a^	0.8 ± 0.04 ^a^	0.8 ± 0.04 ^a^	0.5 ± 0.02 ^b^
	Springiness (%)	274.6 ± 75.48 ^a^	251.6 ± 83.73 ^a^	256.9 ± 95.77 ^a^	92.6 ± 1.57 ^b^

^a–d^ Values in the same row with different superscript letters differ significantly (*p* ≤ 0.05).

## Data Availability

Data is contained within the article or Appendix A.

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
