# Peer review of "Enrichment of Wheat Bread with Platycodon grandiflorus Root (PGR) Flour: Rheological Properties and Microstructure of Dough and Physicochemical Characterization of Bread"

_foods, 2023, doi:10.3390/foods12030580_

Round 1
Reviewer 1 Report
The manuscript ID: foods-2159850, entitled "Enrichment of wheat-bread with Platycodon grandiflorus root (PGR) flour: Effect on the rheological properties and microstructure of dough and baking performances, flavor attributes, and nutritional qualities of bread" is very interesting, well written and can be improved if the issues below are addressed.
First all, the authors should improve the article linguistically.
I consider that the title needs to be reformulate. It is too long.
Line 15-16: … flour is considered medical and edible valuable … instead of … flour is considered medical and edible values …
Line 66: … its added value instead of … its added values
Line 77: Why used in the dough bread recipe as ingredients the butter and sugar? The classical recipe includes flour, water, salt and yeast.
Line 80: … 90 mL of distilled water were mixed … How was determine the amount of necessary water?
Line 81: stirring procedure or mixing procedure?
Line 90: Why was used frequency in the range of 1-100 Hz?
Line 90-91: The dough samples without yeast were placed on the 50-mm steel parallel plate with 2.0 mm gap and coated with glycerol …. Please verify the sentence! Probably the edges of the sample were coated with glycerol in order to prevent the moisture loss.
In subsection 2.3, I suggest to include information about loss modulus (G''), loss tangent (tanδ)
Line 125: … bread samples were enriched in a sampling bottle … Please verify the formulation!
Line 161: … gallic acid … instead of … garlic acid …
Line 185: … Tukey’s test instead of … Turkey’s test.
Please include in the Statistical analysis section a briefly describe about OPLS-DA, principal component analysis and hierarchical clustering analysis.
Line 188: … the influence of recipe components … instead of … the influence of components, recipe, …
Line 229-230: … suggesting that the low-stability physical linkage in the network structure ... Please verify the formulation!
Line 233-234: Additionally, the K' value was corresponding to the strength …. Please verify the sentence!
Line 263: … and extensibility …instead of … and extrnsibility …
Line 360: Please write the full name for OPLS-DA and between the parentheses the abbreviation. Abbreviations should be defined the first time they appear.
Line 370: … represented the different correlations … instead of … represented the differential correlation …
References must be numbered in order of appearance in the text and listed individually at the end of the manuscript. Please seen the Instructions for Authors!
Author Response
Thank you for your letter and for the reviewers' comments concerning our manuscript entitled “Enrichment of wheat-bread with Platycodon grandiflorus root (PGR) flour: Effect on the rheological properties and microstructure of dough and baking performances, flavor attributes, and nutritional qualities of bread” (foods-2159850). We sincerely thank you and all reviewers for the critical and valuable feedback. Based on the comments we received, careful revisions have been made to the manuscript and all changes were marked in red font. Please find our point-by-point responses to the reviewers’ and editor’s comments below:

Reviewer 2 Report
Some comments
More details is needed before the review can be carried out - for the electronic nose and tongue profiles. Were there any calibration done? There are limitations where only enriched solutions can be analysed, so what are the pitfalls here?
For ANOVA, missing posthoc information - in Tables for example it contains groupings.
I assume there are some sample replicates, why didn't the author consider replication as an effect here?
Information on OPLSDA is missing, needs to be added.
Each analysis that was carried out should be explained and supplemented with the purpose of it. Right now, it feels like everything is everywhere.
PCA and DFA info is missing.
Push Fig 4 down after electronic nose analysis.
Fig 5. Some form of clustering is also done here, more details needed
Fig 6. It looks like a series of posthoc is ran here, but unsure what, expand.
Author Response

(The authors gave the same response as above.)

Reviewer 3 Report
Dear Authors,
my comments are included in the PDF file (attached).

Author Response

(The authors gave the same response as above.)
